# Health-related quality of life of the Vietnamese during the COVID-19 pandemic

**Mai Quynh Vu**[1☯]*, **Thao Thi Phuong Tran**[1☯], **Thao Anh Hoang**[1☯], **Long Quynh Khuong**[1☯], **Minh Van Hoang**[2☯]

**1** Center for Population Health Sciences, Hanoi University of Public Health, Hanoi, Vietnam, **2** Hanoi University of Public Health, Hanoi, Vietnam

☯ These authors contributed equally to this work.

* vqm@huph.edu.vn

## Abstract

### Background

Vietnam applied strict quarantine measures to mitigate the rapid transmission of the SARS-COV-2 virus. Central questions were how the COVID-19 pandemic affected health-related quality of life (HRQOL) of the Vietnamese general population, and whether there is any difference in HRQOL among people under different quarantine conditions.

### Methods

This cross-sectional study was conducted during 1 April– 30 May 2020 when the COVID-19 pandemic was at its peak in Vietnam. Data was collected via an online survey using Google survey tool. A convenient sampling approach was employed, with participants being sorted into three groups: people who were in government quarantine facilities; people who were under self-isolation at their own place; and the general population who did not need enforced quarantine. The Vietnamese EQ-5D-5L instrument was used to measure HRQOL. Differences in HRQOL among people of isolation groups and their socio-demographic characteristics were statistically tested.

### Results

A final sample was made of 406 people, including 10 persons from government quarantine facilities, 57 persons under self-isolation at private places, and the rest were the general population. The mean EQ-VAS was reported the highest at 90.5 (SD: 7.98) among people in government quarantine facilities, followed by 88.54 (SD: 12.24) among general population and 86.54 (SD 13.69) among people in self-isolation group. The EQ-5D-5L value was reported the highest among general population at 0.95 (SD: 0.07), followed by 0.94 (SD: 0.12) among people in government quarantine facilities, and 0.93 (SD: 0.13) among people who did self-isolation. Overall, most people, at any level, reported having problems with anxiety and/or depression in all groups.

**Data Availability Statement:** All relevant data are within the manuscript and its Supporting information files.

**Funding:** The author(s) received no specific funding for this work.

**Competing interests:** The authors have declared that no competing interests exist.

## Conclusion

While there have been some worries and debates on implementing strict quarantine measures can hinder people's quality of life, Vietnam showed an opposite tendency in people's HRQOL even under the highest level of enforcement in the prevention and control of COVID-19.

## Introduction

The COVID-19 outbreak, being declared a Public Health Emergency of International Concern (PHEIC) by the World Health Organization (WHO), has resulted in many crises on all aspects of the world [1]. According to WHO's COVID-19 dashboard updates, since the first cases were reported in China by January 2020, the pandemic has spread to over 10 million people across the globe, with more than 500 thousand related deaths by 30 June 2020 [2]. In the prevention and control of COVID-19, governments have developed and implemented a wide range of measures including containment and closure (i.e. border closure and entry ban, closing schools, workplaces, public transport, and non-essential businesses, physical distancing and quarantine, as well as limiting public events/gatherings), economic response (i.e. employment and income supports, debt/contract relief for households, and fiscal measures), and health systems (i.e. information campaigns, testing and contact tracing policy, and emergency investment in healthcare) [3]. So aside from being a public health emergency, the COVID-19 pandemic has also directly and indirectly disrupted various socio-economic activities, which in fact, could led to a total of 265 million people (potentially) suffer from starvation and job losses equivalent to 195 million full-time workers [4,5].

Sharing a long border with China at the upper-north region, means that Vietnam can be at higher risk of transmission, yet this developing country has effectively tackled the COVID-19 pandemic. The first case of COVID-19 detected on 23 January 2020 [6]. By 30 June 2020, there has been 355 confirmed cases (of which 335 recovered), with no deaths and no new community-transmitted cases reported since the end of April 2020 [6]. So far, these preliminary successes in the battle against COVID-19 of Vietnam highlighted the highly effective measures in all fronts, including healthcare system, security force, economic policies, along with creative and effective communication campaigns [7]. Most noticeably, Vietnam has been able to stop the rapid transmission of the disease by strictly implementing quarantine measure. In the stage when all COVID-19 cases were ones with travelling records from China and other affected regions (such as Korea and Europe), all citizens entering Vietnam had to comply with a 14-day quarantine via the Official Dispatch No.156/CD-TTg on 2 February [8] the Document No. 1637/BGTVT-VT on 26 February [9] and the Official Dispatch No. 1440/CV-BCĐ on 20 March [10]. In the stage of rapid community transmission of the virus, several 'very-high' risk localities, which were Son Loi-Vinh Phuc, Truc Bach-Hanoi, Ha Loi-Me Linh, and Bach Mai hospital were placed in complete lockdown [11]. During the peak of transmission, a nation-wide physical distancing policy was enacted for the period of 1–15 April 2020 via the Directive No.16/CT-TTg, reflecting the highest level of effort and determination against the pandemic in Vietnam [12]. This practice was then extended until 23 April 2020 [8,13] meaning that more than 95 million people were under self-isolation during the period.

The prevention and control measures against the COVID-19 pandemic in Vietnam aligned and even went beyond WHO's recommendations on physical distancing and precautious quarantine as a core element for limiting the COVID-19 transmission [14]. Nonetheless,

certain impacts of these measures on people's health and well-being could raise some other concerns. In particular, a review revealed that stressors from long quarantine duration, infection fears, frustration, boredom, inadequate supplies and information, financial loss, as well as stigma, could result in post-traumatic stress disorder symptoms, confusion, and anger [15]. Literatures about COVID-19's negative impacts on the health-related quality of life among people with underlying medical condition(s) were also proven elsewhere [16–18]. Therefore, along with economic losses or visible health consequences due to COVID-19, people's health-related quality of life, through the Vietnamese government's disease prevention and control measures, can be foreseen to compromise.

Health related quality of life (HRQOL) is a generic measure, which combines both the quality (health status) and quantity (life years) of health. Several direct methods, for example time trade-off, standard gamble, or rating scales have been developed to quantify HRQOL [19]. Nevertheless a more widely used and less complicated alternative to quantify HRQOL is through a pre-scored multi-attribute health status classification system [19]. The three most commonly used systems are the Health Utility Index (HUI), EQ-5D from the EuroQol Group, and the Short Form 6D (SF-6D) [19]. In Vietnam, the EQ-5D-5L is the first and the only instrument till date to provide HRQOL quantified measurement based on Vietnamese' preference [20]. The HRQOL of Vietnamese general population, measuring by EQ-5D-5L, has been verified [20,21]. A study in 2017 on HRQOL of the Vietnamese, however applying Chinese' preferences, reported a mean EQ-5D-5L value of 0.91 and EQ-VAS score of 87.4 [21]. In the earlier this year, study on HRQOL reference data using Vietnamese EQ-5D-5L showed a slightly lower HRQOL of the Vietnamese at 80.10 and 0.94 for EQ-VAS and EQ-5D-5L value respectively [22]. Considering such findings belong to pre-COVID period, the people HRQOL perhaps would show a decline in the pandemic time.

Evidence on the impacts of the COVID-19 crisis itself and government responses on the general population's HRQOL, especially in Vietnam, is limited to date. It can be hypothesized that, strict mobility restrictions, difficulties in usual and daily activities, along with negative physical and psychological impacts resulted from quarantine or physical distancing, may lead to lower quality of life. Assessing HRQOL within the context of Vietnamese government responding to the COVID-19 pandemic can provide insights into the true effectiveness and inclusiveness of the prescribed policies. Central questions are how COVID-19 affected HRQOL of the Vietnamese general population, and whether there was any difference in HRQOL among people under different quarantine conditions. This study aims to capture patterns of HRQOL among three groups of (1) people being in government quarantine facilities, (2) people being self-quarantined compulsorily at home, and (3) the general population under the government responses towards the COVID-19 pandemic.

## Methods

### Study sample

A convenient sampling approach was applied to recruit participants aged 18 years and above. Participants were sorted into three groups based on their level of contact with suspected/ confirmed cases of COVID-19. The first group included people who had close contact with an infected person, defined as F1. The second group in our analysis included people who had close contacts with these F1 (suspected cases), as F2, F3, and F4. The other people who were not F1-F4 were grouped into the third group as "do not need isolation" in this study. Definitions of "close contact", ranging from the closest (F1) to most distant (F4) were defined by the guideline of the Vietnam Ministry of Health for COVID-19 protection [23]. In short, close contacts were traced and identified as people who shared the same house/residence area,

workplace, tourism group, airplane, or had face-to-face conversation with less than a 2-meter distance. Those of F1 contacts were required to leave their home and be quarantined at government facilities. And so on with F2 contacts were people who had close contacts with suspected cases (F1), they were compelled to be self-isolation at their private place. For people who had close contacts with F2, identified as F3 and F4, were suggested to practice self-isolation at their private place.

## Data collection

A cross-sectional study was conducted during 1 April– 30 May, 2020 when the COVID-19 pandemic was at its peak in Vietnam. On 1st June 2020, as Viet Nam has officially announced being free from community-transmitted cases and having social activities back to normal, the data collection was stopped by early-June to avoid information bias. Data was collected online based using Google Forms, and participants were invited to self-answer the online survey. Google Forms is a survey administration software that allows collecting information from respondents through surveys. The collected information can be automatically entered into a spreadsheet for data extraction [24]. The survey was shared publicly on social media platforms. First, people were given general information of this study and then asked for consent to share their data to the research team. Participants were then directed to a questionnaire with four main sections. The first section contained questions to collect participants' demographic information (such as sex, age, residences, income, medical history and so on). The second one was questions to sort participants into the three above-mentioned groups (F1 group, F2-3-4 group, and "do not need isolation" groups). Participants were asked to claim whether they were currently in government quarantine facilities (F1) or in isolated residence areas (F2-3-4). It was assumed that there could be of "self-isolated at private places" cases (F2-3-4), whom had not yet assigned by the MOH for strict quarantine, and was not sure about their F2-3-4 status. Therefore, sorting questions were applied to determine whether the respondent: (1) shared either living residence or workplace or travel group with the suspected cases; (2) were in the same row or had a seat in two rows from the suspected cases in public transport; and/or (3) had face-to-face conversation with the suspected cases within a distance of less than two meters. Participants, who reported having any of these three experiences, were categorized into the "self-isolation at private place" group. The third section was for participants to self-evaluate their compliance in Likert-scale questions from not doing to comprehensive compliance as in the MOH's guideline for protecting people from COVID-19. Participants were asked to self-assess their compliance with four criteria on physical distancing according to the MOH's guideline, including (1) maintaining distance at 1.8 meters, (2) in-person socialize pauses, (3) travel pauses, (4) direct physical contact avoidance. In case participants were claimed/sorted in the self-isolation group (F2-F4), they were additionally asked to self-assess their compliance with seven criteria on quarantine practice according to the MOH's self-isolation guideline. These seven criteria were (1) completely home-based self-isolation, (2) keeping hygiene of the isolation places, (3) daily temperature check-up, (4) reporting suspected symptoms, (5) physical contact limitation, (6) individual hygiene, and (7) trash classifications. The last part of the online questionnaire was to measure participants' HRQOL during the COVID-19 outbreak using the Vietnamese EQ-5D-5L.

## The EQ-5D-5L instrument

The EQ-5D-5L questionnaire comprises a descriptive system and a visual analogue scale (EQ-VAS). The descriptive system classifies health into five dimensions: mobility, self-care, usual activities, pain/discomfort, and anxiety/depression [25,26]. Within each dimension,

respondents are asked to describe their perceived current health status using five levels of severity (having no problems, having slight problems, having moderate problems, severe problems, and unable to/ having extreme problems) [27,28]. The EQ-VAS was a hash-marked scale ranging from 0 to 100 in which 0 represents the worst imaginable health and 100 for the best imaginable health. The Vietnam EQ-5D value set was applied to calculate values of all health states generated by EQ-5D-5L [20].

## Analysis

Two main outcomes in this study were the EQ-VAS and EQ-5D-5L values. The EQ-VAS scores were attained directly from the participants, and the EQ-5D-5L values representing HRQOL quantification were calculated. Descriptive statistics of the five dimensions five levels, EQ-VAS, and the EQ-5D-5L values were stratified into three groups of participants, including who (1) were in government quarantine facilities, or (2) were self-isolated at their private place or (3) did not need isolation (referred to whom not F1-4). Of which, percentages of the five dimensions and five levels were presented, respectively. Since the sample size of the two groups: "In Government quarantine facilities" and "Self-isolation at private place" were small, and the score of EQ-5D-5L and EQ-VAS were not normally distributed, the Mann–Whitney U tests and Kruskal Wallis tests were carried out to identify the difference in HRQOL among participants. The Dunn's tests were used as the Poc-hoc analysis of the Kruskal-Wallis to examine the differences among multiple pairwise comparisons. The hypothesis was set as HRQOL would be lower among people having worse living conditions, with the lowest expected to be among the group in government quarantine facilities (F1), followed by the group of self-isolation (F2-3-4), and the highest to be among the remaining general population. Kruskal Wallis test was employed to test the hypothesis. A significance level of 0.05 was used for all statistical tests. Data was analyzed using STATA version 15 software (StataCorp, College Station, TX, USA).

## Ethical considerations

The beginning of the survey provided general information of the study, and clearly stated that as an individual proceeded with answering the questions, he or she would have given informed consent to participate in the study and understood the right to withdraw this consent at any time by simply exiting the survey Google Form. The study design was approved by the Ethical Review Board for Biomedical Research at Hanoi University of Public Health (Identification number: 144/2020/YTCC-HD3).

## Results

Table 1 shows the study participants' characteristics. Overall, 409 people responded, and 406 people completed the survey. The participants' average age was 31.8 years. The majority of participants lived in urban areas (71.4%), were females (69.7%), had high education level (92.3% having university and higher education), were working people (73.6%), were of Kinh ethnicity (95.6%), had no religion (93.1%), and not living with any chronic diseases (80.8%). Multi-member families were the most common family type among these participants (77.3%), and incomes were distributed similarly among participants. Overall, most participants were sorted into the "do not need isolation" group (83.5%) as they had no close contact with suspected cases. We found 14.0% of the participants (57 persons) had close contact with suspected cases, which then were sorted into the "Self-isolation" group (F2-3-4). Ten people reported that they were in government quarantine facilities (F1 group).

**Table 1. The demographic and socioeconomic characteristics of the sample.**

|  | n (%) |
|---|---|
| N | 406 |
| Age, mean (SD) | 31.8 (10.7) |
| **Sex** | |
| Male | 123 (30.3) |
| Female | 283 (69.7) |
| **Residence areas** | |
| Rural | 67 (16.5) |
| Urban | 290 (71.4) |
| Suburban | 49 (12.1) |
| **Educational level** | |
| Lower than university | 31 (7.6) |
| University | 275 (67.7) |
| Higher than university | 100 (24.6) |
| **Monthly income of household** | |
| <5 million VND | 62 (15.6) |
| 5–10 million VND | 110 (27.7) |
| 10–20 million VND | 127 (32.0) |
| >20 million VND | 98 (24.7) |
| **Family type** | |
| Living alone | 56 (13.8) |
| Couples | 36 (8.9) |
| Others (Nuclear or Extended family) | 314 (77.3) |
| **Currently have a paid job** | 299 (73.6) |
| **Major ethnicity (Kinh)** | 388 (95.6) |
| **Having no religion** | 378 (93.1) |
| **Currently living with chronic disease(s)** | 78 (19.2) |
| **Isolation status**[*] | |
| Do not need isolation | 339 (83.5) |
| Quarantine in Government quarantine facility (F1) | 10 (2.5) |
| Need to self-isolation at private place (F2-F4) | 57 (14.0) |

Notes:

[*]: According to the Vietnam Ministry of Health, people who has close contact with a COVID-19 patient is needed to follow the quarantine at Government facility (F1). People who has close contact with a COVID-19 suspicious case (F1) is obligated to implementing self-isolation at their private place (F2). For anyone who has close contact with F2 is recommended to do self-isolation (F3/F4). Close contacts are considered as sharing the same residence/apartment/ building/plane/tourism group or having physical contact (within 2 meters) with a suspicious case.

Fig 1 is about the physical distancing compliance among 406 participants during COVID-19 in Viet Nam. The number of people completely followed the in-person event pauses, physical contact avoidance and travel pauses were 297, 259 and 206 persons, respectively. The number of people who comprehensively practiced the 1.8 meters distancing was low at 155 persons. The numbers of people did not practice physical distancing according to MOH's guidance were low but still existed. There were 3 persons reported not doing the 1.8 meters distancing, not practicing in-person event pauses, or not following the travel pauses. Only a person from the general population group reported not doing all four physical distancing guidance.

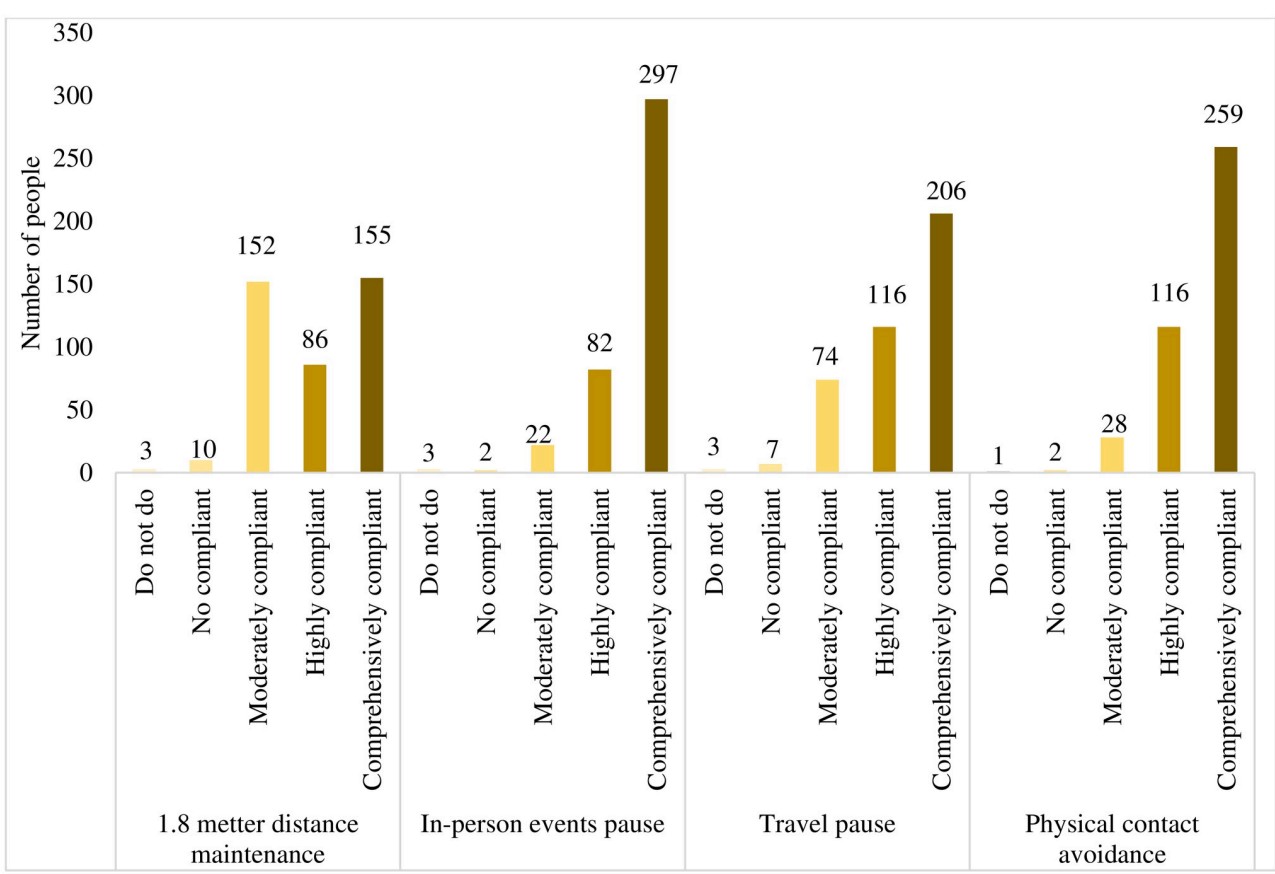

**Fig 1. Number of people complied to "physical distancing" during the COVID-19 in Viet Nam (N = 406).**

Fig 2 shows the percentages of compliance levels towards the MOH's guideline on self-isolation at individual private place. All 57 persons (14.0%) of the "self-isolation" group (identified F2-3-4) responded to this compliance self-evaluation. These people were more likely to comprehensively comply to 14-days of complete self-isolation (61.4%), reporting suspected symptoms (70.2%), limiting physical contacts (57.9%), and practicing individual hygiene (57.9%). However, three self-isolation practices reported not being done, existed in keeping hygiene for the isolation places (1.8%), daily body temperature check-ups (7%), and reporting suspected symptoms (1.8%).

Fig 3 describes the EQ-5D five levels five dimensions of all respondents from the three groups through their isolation status. Overall, the rate of people with full health, defined as having no problems at all five dimensions, was the highest among people in government quarantine facilities (60.0%), followed by 57.9% and 54.9% among groups of people under self-isolation, and not needing isolation, respectively. Most people reported having problems (at any levels) with anxiety/depression, which were 40.1%, 38.6% and 30.0% among people in groups of do not need isolation, self-isolation, and government quarantine facilities, respectively. Mobility was the second highest dimension that participants responded having problems at all levels. All respondents reported having no problems with self-care, except for 3.5% of people in the self-isolation group. Having problems with usual activities and having pain/discomfort were not reported among people in the government quarantine group. However, these dimensions were both reported to be lower at 6.5% among people do not need to isolation, yet were respectively higher at 8.8% and 14.0% among people in the self-isolation group.

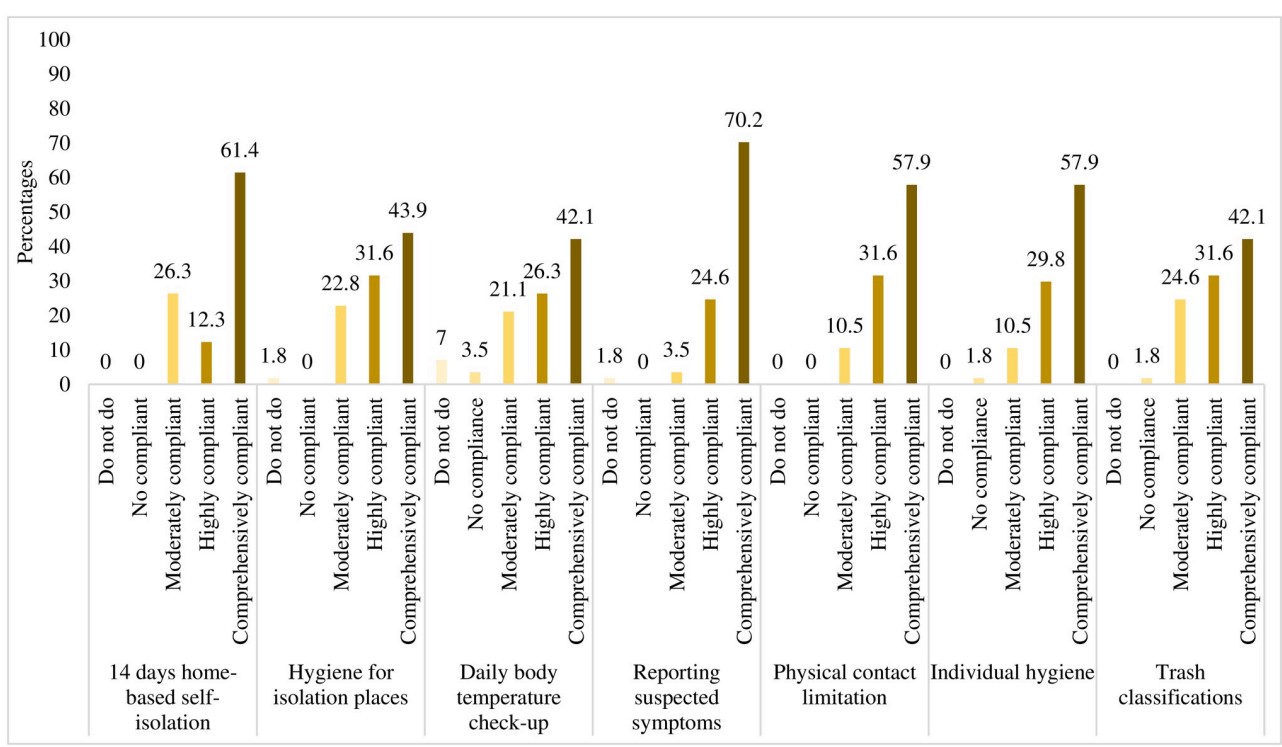

**Fig 2. Percentages of people complied to "self- isolation" during the COVID-19 in Viet Nam (N = 57).**

Table 2 shows means and standard deviations of the EQ-VAS by the three participant groups and their demographic characteristics. Overall, people who were at higher ages, females, and living with chronic diseases reported lower EQ-VAS than the others. As for occupation, whilst currently working people who did not need isolation reported higher EQ-VAS, this was lower for working individuals having to be under self-isolation or in government quarantine facilities. Among the self-isolation group, people who comprehensively complied to the MOH's guidance reported slightly higher EQ-VAS than the others. People who were not in government facilities reporting higher EQ-VAS which also aligned with their higher-level compliance with MOH's guideline on physical distancing. The mean EQ-VAS was reported the highest at 90.50 (SD: 7.98) among people in the government quarantine facilities, followed by 88.54 (SD: 12.24) EQ-VAS in people not needing isolation, and 86.54 (SD 13.69) EQ-VAS in people of the self-isolation group.

Table 3 shows means and standard deviations of the EQ-5D-5L values by the three participant groups and their demographic characteristics. The EQ-5D-5L values were reported lower among people living with chronic disease who were in the groups do not need isolation and self-isolation. The mean EQ-5D-5L value, among people having comprehensively complied to MOH's guidance on self-isolation, was reported to be lower, compared to ones with lower level of compliance. However, for people of the general population who did not need to isolation, the more compliant they were to the MOH's guideline on physical distancing, the higher EQ-5D-5L values reported. Overall, the EQ-5D-5L value was reported the highest among people who did not need to isolation, being at 0.95 (SD: 0.07), followed by 0.94 (SD: 0.12) among people in government quarantine facilities, and 0.93 (SD: 0.13) among people who did self-isolation.

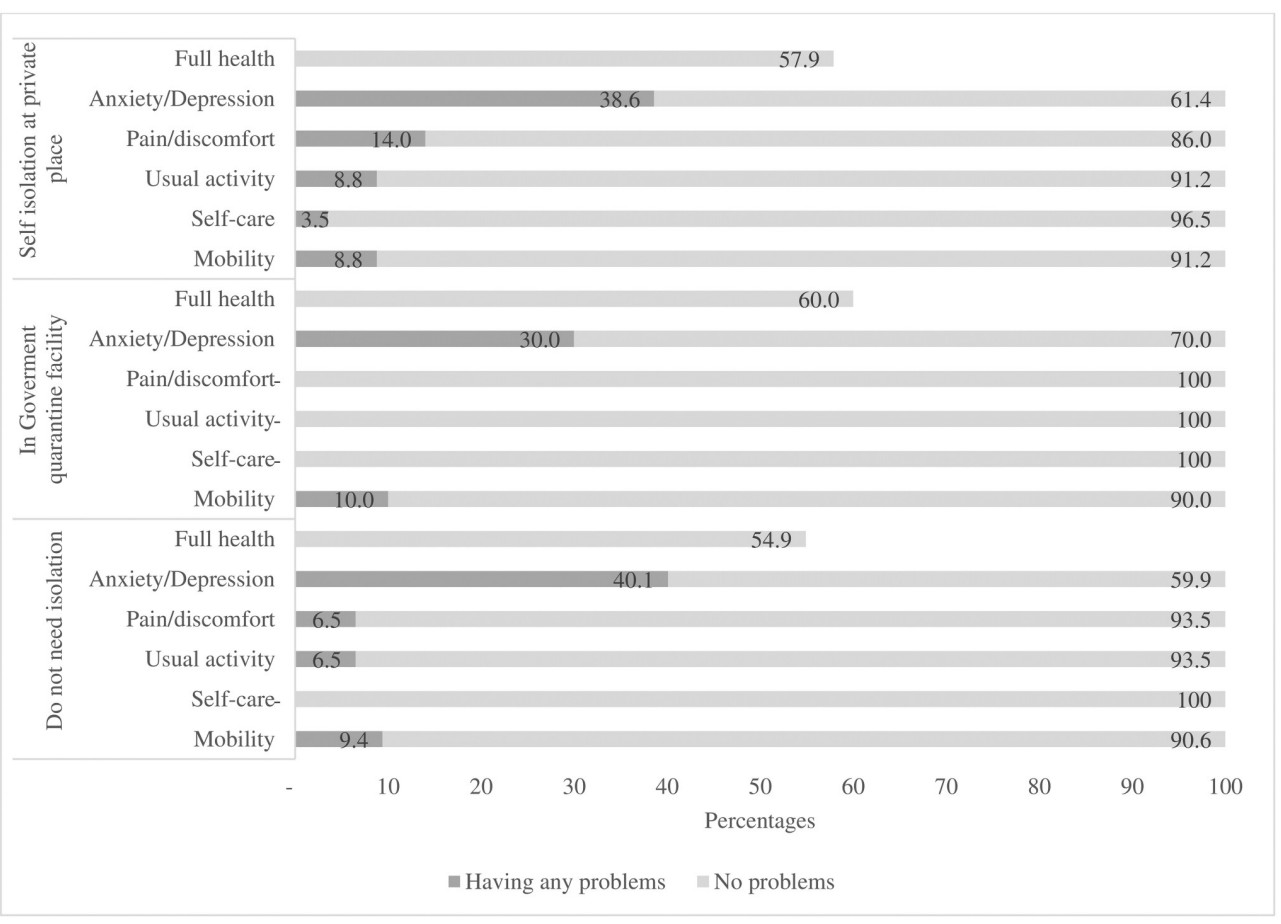

**Fig 3. Patterns of the EQ-5D five dimensions five levels of Vietnamese during COVID-19.**

## Discussion

This study has provided the pattern of the HRQOL for the Vietnamese general population during the COVID-19 pandemic. Differences in HRQOL among the general population not needing compulsory isolation, people in government quarantine facilities, and people implemented self-isolation at private places were given. For the three groups of isolation status, "do not need isolation"; "currently in government quarantine facility" and "self-isolation at private place", the means of EQ-VAS scores reported at 88.54, 90.50, 86.54; and the means of EQ-5D-5L values were 0.95, 0.94, 0.93, respectively. Overall, HRQOL were higher in younger people, and in people with better health and incomes. The percentages of people having full health were higher than 50% among all three groups of different isolation status.

The HRQOL of the Vietnamese during the COVID-19 pandemic reported in this study seemed to be higher than the average HRQOL scores from previous findings. The means of EQ-VAS and EQ-5D-5L values reported in this study were 88.31 and 0.95, respectively. A previous study reported mean EQ-VAS of the general population being at 87.4 and mean EQ-5D-5L values at 0.91 –both were slightly lower than the results found in this study [21]. Nevertheless, the EQ-5D-5L values reported in the previous study were derived from the Chinese 'value set, which means they could not reflect the Vietnamese preferences [21]. In addition, the present study's participants were quite young (average age 31.8 years), having high educational and income levels, which might lead to better HRQOL. Similar patterns between HRQOL and

**Table 2. Mean (standard deviation) of EQ VAS by isolation status during COVID-19.**

| | Do not need isolation | In Government quarantine facilities | Self-isolation at private place |
|---|---|---|---|
| **N** | 339 | 10 | 57 |
| **Age** | | | |
| 18–29 | 87.57 (12.57) | 90.00 (7.07) | 86.48 (12.23) |
| 30–43 | 90.55 (8.77) | 92.50 (9.57) | 87.33 (13.61) |
| 44+ | 88.11 (16.56) | 87.50 (10.61) | 84.89 (18.94) |
| P-value* | 0.13 | 0.74 | 0.75 |
| **Sex** | | | |
| Male | 88.82 (16.92) | 91.67 (10.41) | 87.70 (16.38) |
| Female | 88.42 (9.67) | 90.00 (7.64) | 85.92 (12.21) |
| P-value** | **0.02** | 0.64 | 0.13 |
| **Residence areas** | | | |
| Rural | 88.00 (10.90) | 80.00 (NA) | 88.00 (13.49) |
| Urban | 88.32 (13.01) | 91.67 (7.50) | 87.36 (13.56) |
| Suburban | 90.55 (9.06) | - | 80.00 (15.00) |
| P-value* | 0.55 | 0.22 | 0.39 |
| **Education** | | | |
| Lower than university | 85.33 (18.47) | - | 91.25 (8.54) |
| University | 88.65 (12.32) | 90.71 (7.87) | 84.58 (13.22) |
| Higher than university | 89.30 (8.99) | 90.00 (10.00) | 89.59 (15.38) |
| P-value* | 0.91 | 0.91 | 0.08 |
| **Occupation** | | | |
| Working | 89.19 (10.91) | 88.75 (7.91) | 85.84 (14.82) |
| Not working | 86.82 (15.13) | 97.50 (3.54) | 89.17 (8.21) |
| P-value** | 0.21 | 0.14 | 0.87 |
| **Having any chronic disease(s)** | | | |
| No | 90.06 (9.07) | 90.56 (8.46) | 87.36 (13.49) |
| Yes | 82.71 (19.24) | 90.00 (NA) | 80.71 (14.84) |
| P-value** | **0.00** | NA | 0.11 |
| **Monthly income** | | | |
| <5 million VND | 86.98 (9.82) | 100.00 (NA) | 80.00 (20.98) |
| 5–10 million VND | 89.33 (12.97) | 87.50 (10.61) | 87.00 (7.15) |
| 10–20 million VND | 88.82 (12.58) | 91.00 (7.42) | 89.52 (8.86) |
| >20 million VND | 88.06 (12.82) | 87.50 (10.61) | 84.67 (18.14) |
| P-value* | 0.16 | 0.54 | 0.78 |
| **Compliance to self-isolation** | | | |
| Very compliance | NA | NA | 86.00 (12.28) |
| Completely compliance | NA | NA | 86.06 (12.13) |
| P-value** | | | 0.06 |
| **Compliance to physical distancing** | | | |
| Moderate compliance or less | 80.00 (28.83) | | 76.00 (24.64) |
| Very compliance | 86.39 (14.44) | 95.00 (NA) | 86.25 (6.94) |
| Completely compliance | 90.08 (9.23) | 90.00 (8.29) | 87.53 (13.36) |
| P-value* | **0.05** | 0.60 | 0.47 |
| **Isolation status** | **88.54 (12.24)** | **90.50 (07.98)** | **86.54 (13.69)** |
| K-Wallis—P-value* | | | 0.76 |
| **Overall mean EQ VAS** | | | 88.31 (12.37) |

Notes:

* Results from Kruskal Wallis Tests.

**Results from Mann–Whitney U tests. K-Wallis P-value was from the EQ-VAS difference test among the three participant' groups.

**Table 3. Mean (standard deviation) of EQ-5D-5L value by isolation status during COVID-19.**

| | Do not need isolation | In Government quarantine facilities | Self-isolation at private place |
|---|---|---|---|
| N | 339 | 10 | 57 |
| **Age** | | | |
| 18–29 | 0.95 (0.08) | 0.89 (0.18) | 0.93 (0.10) |
| 30–43 | 0.96 (0.05) | 0.98 (0.03) | 0.97 (0.07) |
| 44+ | 0.95 (0.07) | 0.97 (0.05) | 0.83 (0.25) |
| P-value* | 0.37 | 0.74 | **0.04[a]** |
| **Sex** | | | |
| Male | 0.94 (0.10) | 0.98 (0.04) | 0.92 (0.18) |
| Female | 0.96 (0.06) | 0.93 (0.14) | 0.94 (0.10) |
| P-value** | 0.92 | 0.70 | 0.59 |
| **Residence areas** | | | |
| Rural | 0.95 (0.07) | 0.94 (NA) | 0.95 (0.06) |
| Urban | 0.95 (0.08) | 0.94 (0.12) | 0.93 (0.14) |
| Suburban | 0.96 (0.05) | - | 0.88 (0.13) |
| P-value* | 0.79 | 0.38 | 0.30 |
| **Education** | | | |
| Lower than university | 0.93 (0.11) | - | 0.96 (0.07) |
| University | 0.95 (0.07) | 0.93 (0.14) | 0.93 (0.10) |
| Higher than university | 0.96 (0.05) | 0.98 (0.04) | 0.92 (0.19) |
| P-value* | 0.71 | 0.73 | 0.79 |
| **Occupation** | | | |
| Working | 0.96 (0.06) | 0.98 (0.03) | 0.93 (0.15) |
| Not working | 0.94 (0.09) | 0.81 (0.27) | 0.94 (0.07) |
| P-value** | 0.21 | 0.46 | 0.60 |
| **Having any chronic disease(s)** | | | |
| No | 0.96 (0.06) | 0.94 (0.12) | 0.94 (0.10) |
| Yes | 0.92 (0.10) | 1.00 (NA) | 0.85 (0.28) |
| P-value** | **0.00** | NA | 0.38 |
| **Monthly income** | | | |
| <5 million VND | 0.93 (0.09) | 1.00 (NA) | 0.91 (0.11) |
| 5–10 million VND | 0.95 (0.08) | 0.97 (0.05) | 0.96 (0.06) |
| 10–20 million VND | 0.96 (0.06) | 0.99 (0.03) | 0.95 (0.08) |
| >20 million VND | 0.95 (0.06) | 0.78 (0.22) | 0.9 (0.2) |
| P-value* | 0.46 | 0.29 | 0.68 |
| **Compliance to self-isolation** | | | |
| Very compliance | NA | NA | 0.94 (0.12) |
| Completely compliance | NA | NA | 0.93 (0.10) |
| P-value** | | | 0.59 |
| **Compliance to physical distancing** | | | |
| Moderate compliance or less | 0.9 (0.19) | | 0.78 (0.36) |
| Very compliance | 0.94 (0.08) | 1.00 (NA) | 0.98 (0.03) |
| Completely compliance | 0.96 (0.06) | 0.94 (0.12) | 0.93 (0.10) |
| P-value* | 0.17 | 0.49 | 0.30 |
| **Isolation status** | **0.95 (0.07)** | **0.94 (0.12)** | **0.93 (0.13)** |
| K-Wallis—P-value* | | | 0.93 |
| **Overall mean EQ-5D-5L value** | | | 0.95 (0.08) |

Notes:

* Results from Kruskal Wallis Tests.

**Results from Mann–Whitney U tests. K-Wallis P-value was from the EQ-VAS difference test among the three participant' groups.

[a]Poc-hoc pairwise comparison: p-value "18–29" vs "30–43" = 0.011; p-value "18–29" vs 44+ = 0.222; p-value "30–43"vs 44+ = 0.011.

demographic details of age, educational level, income were found previously [29–31]. Overall, the HRQOL of the general population (group of people not needing isolation) was likely to be higher than both other groups of people in government quarantine facilities and under self-isolation. It appeared that the EQ-5D-5L values was slightly lower among individual being under compulsory quarantine than ones not needing isolation. Such findings aligned with some debates implying that implementing lockdown could ultimately curb the pandemic transmission, but compromise people's mental health [32]. Nevertheless, statistic tests for such HRQOL differences were found insignificant in the present study. So it seemed that strict quarantine measures of COVID-19 prescribed by the Vietnamese government may have not generated negative impacts on people's quality of life. In fact, the HRQOL reduction among people who has close contact with suspected/confirmed cases (from F1 to F4) could be perceived as just a short-term phenomenon, and could be expected to rebound when the virus transmission was well under control.

It is logical that person who had closer contact with the infected cases or suspicious cases may have been confronted with more anxiety or restrictions in doing their usual activities/ mobility/self-care. Therefore, it could contribute to a lower mean of EQ-5D-5L value among person who were in government quarantine facilities and self-isolation at their private place, compared with the those who did not need isolation. Nevertheless, the reported EQ-5D-5L values differences were small, which would be attributed to the related measures and responses taken by the government during the COVID-19 pandemic. According to the Resolution No. 37/NQ-CP on special regime in prevention and control of COVID-19, issued on 29 March 2020 [33] financial support of 80,000 VND/day was given to all people being quarantined at government facilities, and free essential products were also provided (drinking water, face towels, facial masks, water for hand-washing, oral antiseptic, toothbrushes, soap and others). In addition, the government have implemented policies on supporting people being affected by the pandemic issued by the Decision No. 15/2020/QD-TTg [34]. In which, lay-off workers and bankrupted family-business caused by the pandemic, as well as other vulnerable groups like the poor and near-poor households were benefited with some financial means. It can be said that the government in Vietnam has made considerable efforts to mitigate the impacts of strictly quarantine measures, which may help to maintain people's quality of life during their time of complete quarantine or in-place self-isolation.

The present study found that people who aged 44 and above, and people living with chronic diseases often reported lower HRQOL than ones without. This HRQOL pattern were also found in other previous findings [30,31,35]. Moreover, the HRQOL differences by ages and chronic disease status in the self-isolation group were the most visible, yet these were less noticeable among the general population group who did not need to be self-isolated. This is understandable since old people and individuals with underlying medical conditions were shown to be more likely to have severe illness if being infected with COVID-19 [36]. Specifically, potential underlying conditions coming with ageing, could also be a "double attack" placing old people at even higher risk of the disease [37]. Therefore, results from this study also implied the essence to take better care for the elderly and individuals living with chronic diseases during the pandemic.

When the physical distancing strategy was implemented nation-wide in Vietnam, we conducted the data collection timely to capture the HRQOL of the Vietnamese population during this period. However, this could be considered a double-edge sword and had caused several limitations to the study. First, the data was collected based on an online survey which could be prone to selection bias. Due to the nature of accessing internet and technology was, in fact, less common for the elderly and ones with low socio-economic status, most of our participants were young and from the middle class. The elderly, poor and other vulnerable groups did not

seem to appear in our sample, and so their HRQOL status was not able to be recorded. Secondly, the sample size of this study was rather small, with only 10 people being in government quarantine facilities, and 57 persons having to implement self-isolation, the power of the statistical tests, therefore, might not be sufficient. Therefore, this sample could not be large enough to reflect the whole picture for HRQOL of the Vietnamese general population during COVID-19. Nonetheless, a national lockdown has been a huge barrier for all research to obtain data in the most desirable, wholesome manner, not only for our study.

## Conclusion

With respect to all the perceived worries and controversies about the potential of hindering quality of life due to strict quarantine measures, our findings in Vietnam has made it a good case study for successfully implementing those measures at the high level of enforcement, without affecting people's HRQOL.

## Supporting information

**S1 Data.**
(XLS)

## Acknowledgments

The authors are grateful to all respondents who had spending time on the survey. The authors thank Google LCC for free sharing the online survey tool, which were utilized in this study. We would also like to thank our colleagues at Center for Population Health Sciences for helping to introduce the survey publicly.

## Author Contributions

**Conceptualization:** Mai Quynh Vu, Minh Van Hoang.

**Data curation:** Thao Anh Hoang, Long Quynh Khuong.

**Formal analysis:** Mai Quynh Vu, Thao Anh Hoang, Long Quynh Khuong.

**Investigation:** Mai Quynh Vu.

**Methodology:** Mai Quynh Vu, Thao Anh Hoang, Long Quynh Khuong.

**Project administration:** Mai Quynh Vu, Thao Thi Phuong Tran.

**Software:** Mai Quynh Vu, Long Quynh Khuong.

**Supervision:** Mai Quynh Vu, Minh Van Hoang.

**Validation:** Mai Quynh Vu, Minh Van Hoang.

**Visualization:** Mai Quynh Vu.

**Writing – original draft:** Mai Quynh Vu, Thao Thi Phuong Tran, Thao Anh Hoang.

**Writing – review & editing:** Mai Quynh Vu, Thao Thi Phuong Tran, Thao Anh Hoang, Minh Van Hoang.

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
