## [Decision Letter · Decision Letter 0]

14 Oct 2020

PONE-D-20-22842

HEALTH-RELATED QUALITY OF LIFE OF THE VIETNAMESE DURING THE COVID-19 PANDEMIC

PLOS ONE

Dear Quynh Mai Vu,

Thank you for submitting your manuscript to PLOS ONE. After careful consideration, we feel that it has merit but does not fully meet PLOS ONE’s publication criteria as it currently stands. Therefore, we invite you to submit a revised version of the manuscript that addresses the points raised during the review process.

Please submit your revised manuscript on or before 13 November 2020. If you will need more time than this to complete your revisions, please reply to this message or contact the journal office at plosone@plos.org. Please include the following items when submitting your revised manuscript:

We look forward to receiving your revised manuscript.

Kind regards,

Olanrewaju Oladimeji, Ph.D., MB; BS

Academic Editor

PLOS ONE

Journal Requirements:

"Human Subject Research (involving human participants and/or tissue)

- Name of the institutional review board or ethics committee that approved the study: The study designs were considered and approved by the Ethical Review Board for Biomedical Research at the Hanoi University of Public Health.

- Approval number and/or a statement indicating approval of this research: 144/2020/YTCC-HD3".   

a. Please provide additional details regarding participant consent. In the ethics statement in the Methods and online submission information, please ensure that you have specified what type you obtained (for instance, written or verbal, and if verbal, how it was documented and witnessed).

If your study included minors, state whether you obtained consent from parents or guardians.

If the need for consent was waived by the ethics committee, please include this information.

3. Please include your tables as part of your main manuscript and remove the individual files. Please note that supplementary tables should be uploaded as separate "supporting information" files.

4. Your ethics statement should only appear in the Methods section of your manuscript.

If your ethics statement is written in any section besides the Methods, please move it to the Methods section and delete it from any other section.

Please ensure that your ethics statement is included in your manuscript, as the ethics statement entered into the online submission form will not be published alongside your manuscript.

Reviewers' comments:

Reviewer's Responses to Questions

**Comments to the Author**

1. Is the manuscript technically sound, and do the data support the conclusions?

Reviewer #1: Yes

Reviewer #2: Partly

2. Has the statistical analysis been performed appropriately and rigorously? 

Reviewer #1: Yes

Reviewer #2: Yes

3. Have the authors made all data underlying the findings in their manuscript fully available?

Reviewer #1: No

Reviewer #2: Yes

4. Is the manuscript presented in an intelligible fashion and written in standard English?

Reviewer #1: Yes

Reviewer #2: Yes

5. Review Comments to the Author

Reviewer #1: This study assessed Health related Quality of Life (HRQOL) among three groups (Isolation at Government facility, Self-isolation and general population not isolated) of Vietnamese population at the peak of the strict COVID quarantine measures between April and May 2020. Data was collected virtually using Google tool. EuroQual Vietnamese validated version was used for data collection. The study concluded that contrary to expectation, the HRQOL of the three groups of Vietnamese was better during the strict quarantine measure than was before and adduced this to better palliative measures by the Government.

General Comments:

The manuscript contribute to body of scientific knowledge by providing insight into the living condition and health status of the Vietnamese population during COVID quarantine measure. However, the sample size is rather low for generalization to a population of over 90 Million. Although the authors expressed this as one of the limitations of the study, rather a sample size of 10 for the Government quarantine facility is a major flaw to the study.

Introduction:

The syntax errors in lines 40 and line 57 should be corrected. prepositions are missing

Literature review lacked HRQOL among the Vietnamese population pre-COVID. The authors should include a few sentences on this.

Methods:

Line 82-83: How does ending data collection process at the end of strict lock-down prevent information bias?

Line 83: Google survey tool should be properly described and reference

Line 87: Use of "e.g" and "etc" should be discouraged in biomedical writing

Line 90-93: The use of sort questions to classify participants into the three groups is unnecessary, reason being that the grouping in this study can only be objectively informed by the living condition during the quarantine policy rather their COVID status.

Line 95-102: Can compliance to the quarantine policy be a major confounder in this study?

Line 106 -108: These statements require proper references

Analysis:

Line 120-121: The assumptions for applying Mann-Whitney U and Kruskal Wallis tests should be clearly stated in addition to their limitations

No statement on significance level was made. This is a vital to the study

Results:

In general, the authors should clearly state significant findings in this study

Table 1: Shows a very young population (31.8 years). Could this have confounding influence on HRQOL and quarantine? This worth exploring in the analysis by controlling for the age. This was corroborated with an interesting finding in Table 2 where 44+ years have reduced HRQOL compared to younger age groups

Table 2: With only 10 participants in (F1 group) the group that was hypothesized to have worse HRQOL, how reliable is this?. Furthermore, the Kruskal Wallis test also show too many empty cells and therefore make some of the analysis unreliable

Tables 2 & 3: Kruskal Wallis test is an Omnibus test and therefore require a Poc hoc analysis to reflect the significant combinations. The authors should include this in their analysis

Discussion:

Line 191-194: The main findings of HRQOL among the three groups studied were statistically insignificant from the results reported but the author based the main conclusion on this findings. It will be desirable if the authors can include other salient significant findings from the inferential statistics in both the discussion and the summary (e.g Compliance, age, sex, chronic illness). Critically appraisal of these factors will give more credence to the study

References:

All the internet references lacked accessed date. This should be indicated in the relevant

Reviewer #2: This manuscript assessed the HRQOL of the Vietnamese during the lockdown instituted in response to the Covid – 19 pandemic. It was able to demonstrate that the quarantine measures instituted by the Vietnamese government had not reduced the HRQOL of the general population. There are however minor issues that may require correction and clarification as follows:

1. Limitation of the sample size has been addressed in your discussions, however the composition of your sample and its effect on your final outcome may require further discussion/clarification owing to the association of higher HRQOL with higher education levels [I, II] and 92.3% of your respondents having university degrees and above.

References

I. Hoi, Le & Chuc, Nguyen & Lindholm, Lars. (2010). Health-related quality of life, and its determinants, among older people in rural Vietnam. BMC public health. 10. 549. 10.1186/1471-2458-10-549.

II. Mielck, Andreas ,Reitmeir, Peter, Vogelmann, Martin, Leidl, Reiner(2012)

Impact of educational level on health-related quality of life (HRQL): results from Germany based on the EuroQol 5D (EQ-5D) European Journal of Public Health.

DO 10.1093/eurpub/ckr206

III. Gil-Lacruz, M., Gil-Lacruz, A.I. & Gracia-Pérez, M.L. Health-related quality of life in young people: the importance of education. Health Qual Life Outcomes 18, 187 (2020). https://doi.org/10.1186/s12955-020-01446-5

Other minor issues include;

2. Line 138 - Data set incorrectly ordered for the dimension character listed prior and its corresponding numerical value. The correct order should be 297, 259, and 206 respectively (as opposed to 297, 206 and 259).

3. Line 166 – Table 2 should read mean and standard deviation (as opposed to median and standard deviation)

4. Line 161-163 report of result may require restructuring of sentence for clarity.

Otherwise the paper is clear, concise and informative.

6. PLOS authors have the option to publish the peer review history of their article (what does this mean?). If published, this will include your full peer review and any attached files.

Reviewer #1: **Yes: **Oladimeji Akeem Bolarinwa

Reviewer #2: No

---

## [Author Response · Author response to Decision Letter 0]

12 Nov 2020

Enclosed file: Responses to reviewers

---

## [Editor Report · Decision Letter 1]

7 Dec 2020

Health-related Quality of Life of the Vietnamese during the COVID-19 Pandemic

PONE-D-20-22842R1

Dear Dr. Mai Quynh Vu,

We’re pleased to inform you that your manuscript has been judged scientifically suitable for publication and will be formally accepted for publication once it meets all outstanding technical requirements.

Kind regards,

Olanrewaju Oladimeji, Ph.D., MB; BS

Academic Editor

PLOS ONE
---

## [Editor Report · Acceptance letter]

9 Dec 2020

PONE-D-20-22842R1 

Health-related Quality of Life of the Vietnameseduring the COVID-19 Pandemic 

Dear Dr. Vu:

I'm pleased to inform you that your manuscript has been deemed suitable for publication in PLOS ONE. Congratulations! Your manuscript is now with our production department. 

Kind regards, 

on behalf of

Dr. Olanrewaju Oladimeji 

Academic Editor

PLOS ONE